# Impact of Nanoclay on the pH-Responsiveness and Biodegradable Behavior of Biopolymer-Based Nanocomposite Hydrogels

**DOI:** 10.3390/gels5040044

**Published:** 2019-10-16

**Authors:** Arti Vashist, Anujit Ghosal, Atul Vashist, Ajeet Kaushik, Y. K. Gupta, Madhavan Nair, Sharif Ahmad

**Affiliations:** 1Materials Research Laboratory, Department of Chemistry, Jamia Millia Islamia, New Delhi 110025, India; avashist@fiu.edu (A.V.); anuj.ghosal@gmail.com (A.G.); 2Center for Personalized Nanomedicine, Institute of NeuroImmune Pharmacology, Department of Immunology & Nanomedicine, Herbert Wertheim College of Medicine, Florida International University, Miami, FL 33199, USA; akaushik@fiu.edu (A.K.); nairm@fiu.edu (M.N.); 3School of Lifesciences, Beijing Institute of Technology, Beijing 100081, China; 4Department of Biotechnology, All India Institute of Medical Sciences, New Delhi 110029, India; atulvashist1980@gmail.com; 5Department of Natural Sciences, Division of Sciences, Art, & Mathematics, Florida Polytechnic University, Lakeland, FL 33805-8531, USA; 6Department of Pharmacology, All India Institute of Medical Sciences, New Delhi 110029, India; yk.ykgupta@gmail.com

**Keywords:** nanocomposite hydrogels, Cloisite 30B, drug delivery system, chitosan, guar gum, biodegradability, pH-responsive

## Abstract

This research work deployed free radical polymerization for the development of pH-responsive hybrid nanocomposite hydrogels (NCHs) with the formation of improved interpenetrating networks (IPN). The crosslinked biopolymeric system was composed of (chitosan (CH)/guar gum (GG)/polyol) and a nanofiller (Cloisite 30B). The study was aimed to investigate the role of Cloisite 30B as a nanofiller and linseed oil-derived polyol to induce stable interpenetrating networks in chitosan‒guar gum-based hydrogels. FT-IR analysis confirmed the formation of crosslinked networks with the formation of hydrogen bonds in the synthesized NCHs. Thermogravimetric analysis and differential scanning calorimetry revealed high thermal stability of the NCHs. The hydrolytic and soil burial degradation tests confirmed the biodegradability of the synthesized NCHs. An extraordinarily high swelling capacity in a buffer solution of pH 4.0 and 7.4 demonstrated their pH-responsive behavior. It has been demonstrated that even the minimal addition of polyol to the guar gum-based hydrogels has influenced the stability and characteristic features such as high swelling capacity owing to the formation of interpenetrating networks and the biodegradability of the hydrogels.

## 1. Introduction

Hydrogels are hydrophilic polymers with three-dimensional structures and a superabsorbent property, showing the uptake of a large number of fluids [1,2]. Hydrogels are responsive to various stimuli like light, temperature, pH, electric current, and magnetic field, which makes them a suitable candidate for drug delivery systems (DDS) [3,4,5,6,7]. The combination of natural polymers with nanofillers and bioactive gives rise to stable nanocomposite hydrogels (NCHs) [8,9,10]. Inspired by the completely new properties and applications of hydrogels, which arise due to the addition of nanoforms such as metal nanoparticles [11], clay [12], silica [13], and graphene [14], the NCHs have emerged as next-generation materials and have potential application as biomedical devices [15], biosensors [16], hygiene [17], wastewater treatments [18], agricultural applications [19] drug release [20], tissue engineering [21], scaffolds [22], and smart materials [23]. 

Hydrogel technology is a rapidly growing field and finds several applications in biomedical applications. The need the hydrogels derived from natural resources has gained enormous attention in recent years since they are biodegradable and have the potential to be used for in vivo applications [24]. Guar gum (GG) is known as a hydrophilic natural non-ionic polysaccharide, safe and non-toxic, and has been extensively explored for drug delivery. It is a branched biopolymer with β-d-mannopyranosyl units, which contributes to its superabsorbent property [25]. The ability to vary its rigidity and the nontoxicity are reasons for its use as a vehicle for oral [26], specifically colon systems and in the large intestine [27]. Chitosan (CH), an excellent biopolymer with unique characteristics of biodegradability, biocompatibility, hydrophilicity, and the presence of cationic charge, is the best candidate for developing a biocompatible DDS [28,29]. Linseed polyol, a sustainable, resource-based compound, was used for the hydrophobic modification [30]. Thus, in the present study, natural polymers like chitosan and guar gum were used to develop novel stable NCHs.

Recently, an innovative approach of using clay minerals as filler in a hydrogel matrix has come up with an easy and biofriendly route of synthesis of DDSs. The different types of clay used include natural silicate, vermiculite, montmorillonite, hectorite, etc. and synthetic silicates (laponite and flurorectorite). Cloisite 30B is one of the commonly used nanoclays [31]. The exfoliated silicate layers of cloisite 30B have polar functional groups and are compatible with the polymers having polar groups. Clay minerals possess environmentally friendly characteristics and are generally employed for the designing of superabsorbent biomaterials [32,33]. Wang et al. used guar gum and organic nanoclay such as organic rectorite CTA+ to fabricate superabsorbent nanocomposites with a high swelling capacity [19]. Liu et al. developed alginate/quaternized carboxymethyl chitosan/clay nanocomposite microspheres and used them as a controlled release carrier [34]. These microspheres containing organic clay showed superior encapsulation and release capacities for bovine serum albumin. In our earlier studies, we have explored the effect of the linseed oil-based polyol using natural polymer hydroxyl ethyl cellulose (HEC) [35], as well as synthetic methyl methacrylate (MMA) [36]. Polyol seems to dominantly affect all the characteristic features by introducing interpenetrating networks.

In this study, we have synthesized and characterized the Cloisite 30B dispersed nanocomposite hydrogels using the biopolymers chitosan and guar gum. Structural, thermal, and morphological studies were carried out using Fourier-transform infrared (FT-IR), thermal gravimetric analysis (TGA), differential scanning calorimetry (DSC), scanning electron microscopy (SEM), and transmission electron microscopy (TEM). The present study aimed to exploit the role of Cloisite 30B as a nanofiller and linseed oil-derived polyol to induce stable interpenetrating networks in chitosan‒guar gum-based hydrogels. The highlight of the study was the synthesis of completely biodegradable and stable hydrogels, which were hydrophobically modified using a sustainable resource-based polyol and nanofiller cloisite 30B. This is a new method of modification for hydrogels and for the introduction of interpenetrating networks, which resulted in hydrogels with high swelling and the desired functionality. 

## 2. Results and Discussion

### 2.1. Synthesis of Nanocomposite Hydrogels

A free radical polymerization method was used to synthesize the nanocomposite hydrogels. Biodegradable chitosan and guar gum polymers were used to develop the hydrogel matrix. Two sets of compositions without Cloisite 30B (CH‒GG/polyol) (Table 1) and with the nanofiller Cloisite 30B (NCHs) (Table 2) were prepared. To study the impact of hydrophobic modification and the nanofiller, different concentrations of polyol and cloisite 30B were added to the feed compositions. The polyol acted as a crosslinking agent to form the interpenetrating 3D network, which was further improved by adding Cloisite 30B. It was observed that both polyol and cloisite 30B played a major role in the stability, crosslinked structure, swelling characteristics, and biodegradability of hydrogels, as is discussed in detail in the next sections.

### 2.2. Structure and Stability of Hydrogels

The reaction scheme showing the mechanism of formation of CH‒guar gum‒polyol hydrogels is presented in Figure 1. 

### 2.3. FT-IR Spectra of NCHs

The FT-IR absorption spectra of chitosan, CG-0, and chitosan‒guar gum‒polyol (CGP2-1 to 5), NC-2, NC-4, and NC-6 samples are shown in Figure 2. The FT-IR spectra of guar gum show a characteristic broad and strong peak at 3391 cm^−1^, which may be attributed to the OH bond stretching. The sharp band at 2907 cm^−1^ may be attributed to the CH group stretching. CH_2_ group bending is observed in the band appearing at 1457 cm^−1^ and CH_2_–O–CH_2_ bending appears in the 1025 cm^−1^ frequency region [37,38]. The peak at 1090 cm^−1^ is assigned to the ether linkages in chitosan. 

The peak at 1442 cm^−1^ represents the characteristic amino peak and the peak at 1660 cm^−1^ is for amide I, which is attributed to the acetamide group of chitosan [39], as shown in Figure 2. It is clearly seen that there is intermolecular hydrogen bonding between the –OH groups present on the chitosan and guar gum and the –OH groups of the polyol and Cloisite 30B. The broadening of the band near 3000–3200 cm^−1^ is also observed in the case of CGP2-3 and CGP2-5 and NC-4, but it is not as broad as seen in NC-6 (marked with a circle in Figure 2). A wide broadening is observed in the IR spectra for NC-6 for the band in the frequency range 3600–3000 cm^−1^. The peak of the ether group in CGP2-3, CGP2-5, NC-4, and NC-6 appears at 1036 cm^−1^ in the IR spectra, which suggests the formation of, new ether linkages in a hydrogel containing more content of polyol and Cloisite 30B, viz., NC-6, NC-4. However, it was observed that the basic structure of the polymer matrix remained unaffected and only a slight shift in the IR band intensity of the characteristic absorption bands was observed. This may be attributed to the presence of more electrostatic interactions in CGP2-5 (containing more polyol content) as well as the interaction of clay nanoparticles with the polymer matrix in NC-6. 

### 2.4. SEM Analysis

The swollen structures of CG-0 composition (Figure 3a,b) also revealed a crosslinked structure. The presence of guar gum has greatly influenced the crosslinked structure of the matrix. On the other hand, the dried structures in the composition having no polyol content, i.e., CG-0, are not homogeneous and clearly show a rough morphology (Appendix A). The dried surface morphology of the hydrogels is shown in Appendix A for CGP2-3 and in Appendix A for CGP2-5.

The thin pore wall of the hydrogels appears not to be strong enough and excessive agglomeration is seen, which could affect the swelling capacity in the CG-0 hydrogel.

The water-swollen micrographs of CGP2-5 in Figure 4a,b showed a more crosslinked structure than the hydrogels having no polyol (CG-0). The mesh size of IPN hydrogels has increased due to the increased network density. This increase in the mesh size can be attributed to the more hydroxyl groups of polyol, which are able to form hydrogen bonds with chitosan and the guar gum matrix. It should be noted that the porous structure/network observed during SEM analysis may be due to the ice-templating effect. During freezing, the water molecules within the hydrogel network form ice crystals, which on lyophilization leave porous networks.

The SEM images in Figure 5a–c showed three-dimensional interpenetrating network formation following the introduction of Cloisite 30B. The pore dimension or mesh size of the resulting swollen NCHs decreased due to the increase in Cloisite 30B content. This is further supported by the fact that the complete exfoliation of clay minerals and the excess of hydroxyl groups of polyol and clay increased the intermolecular hydrogen bonding, resulting in denser, crosslinked structures.

### 2.5. XRD Analysis

The addition of Cloisite 30B to the CH‒guar gum/polyol matrix in the dispersed form gives rise to strong polar interactions of the hydroxyl groups present in the biopolymeric matrix with the hydroxyl groups of clay particles. The occurrence of hydrogen bonding affected the formation of intercalated and exfoliated structures. The uniform dispersion and exfoliation of clay particles are confirmed by TEM analysis. The XRD spectra of native Cloisite 30B show the presence of a peak at 2θ—4.491° and confirm that the interlayer spacing of the crystalline structure d001 is 19.66 Å [40]. The NC-4 demonstrates a characteristic peak at 20.4° and for NC-6 at 20.6°, as depicted in Figure 6. The diffraction of chitosan shows two different peaks at 2θ = 10° and 2θ = 20° [41]. The peaks at 10° disappeared in NCHs due to the crosslinking reaction. This may be attributed to the destruction of the intermolecular hydrogen bonds between the amine groups and hydroxyl groups of chitosan due to the graft copolymerization. These results indicate that the graft copolymerization caused destruction of the ordered crystal structure of the chitosan. There is an intensity difference between the two compositions, which may be due to the presence of different amounts of Cloisite 30B. The high amount of Cloisite 30B in NC-6 gives rise to a slightly disordered structure in comparison to NC-4, which is also evident from the DSC analysis in which an exothermic peak was observed for NC-4. These profiles indicate an increase in the d-spacing of the resulting NCHs, which implies that Cloisite 30B has been completely intercalated in the CH‒guar gum‒polyol matrix during the polymerization process. 

### 2.6. Thermal Gravimetric Analysis of NCHs

The TGA curves for various NCHs are depicted in Figure 7. The 10% weight loss occurred in the temperature range 200–230 °C, which can be attributed to the evaporation of solvent molecules from the hydrogel network. Half of the weight loss was observed in the range of 380–400 °C, which can be due to the loss of ether and amide linkages, while 70% weight loss occurred in the temperature range 450–480 °C due to the loss of the hydrocarbon chain of linseed polyol [42]. Furthermore, TGA analysis revealed that NC-6 shows less thermal stability in comparison to NC-2 and NC-4. This can be ascribed to the increase in intermolecular interactions as well as the crosslinked network structure between the hydrogel matrix and clay nanoparticles in NC-2 and NC-4. The excess of clay content in NC-6 hydrogels may clog the interstitial spaces of the interpenetrating networks, which may hinder the free rotation of shorter polyether links and result in the low thermal stability of NC-6. The TGA thermograms of CG-0, CGP2-3, and CGP2-5 hydrogels are presented in Appendix A. 

### 2.7. Differential Scanning Calorimetry of NCHs

The DSC thermograms for NC-4 and NC-6 are depicted in Figure 8. The glass transition of NCHs was observed in the temperature range 53–95 °C. Furthermore, a more intense exothermic peak for NC-4 was observed at a slightly lower temperature (225 °C) in comparison to NC-6 (230 °C). This can be ascribed to the formation of an ordered structure due to more ordered interactions and high crosslinking in NC-4 as compared to NC-6, which is also evident from the XRD data. The DSC analysis of CG-0, CGP2-3, CGP2-5, and CGP2-7 is presented in Appendix A. 

### 2.8. Effect on Swelling Mechanism

The combination of a biopolymeric hydrophilic matrix of chitosan and guar gum with the exfoliated cloisite 30B and a hydrophobic covering of vegetable oil-derived polyol merge, forming an interpenetrating nanocomposite hydrogel network exhibiting an excellent water-absorbing capacity. The swelling mechanism behind the synthesized NCHs can be best understood in terms of the interactions occurring between the water molecules and the glassy dried hydrogels. Initially, when water molecules come into contact with the hydrogel network, the polar groups, viz., –OH, –NH_2_, present on the surface of the hydrogels come into play and water penetrates into the polymeric network. The imbibed water can be in the form of free water or bound water, which is also known as nonfreezable water [43]. The variation in size and shape with the change in the external stimuli is the most recognizable characteristic of the synthesized NCHs polymeric hydrogels, which can result in either imbibing or expelling free water. 

#### 2.8.1. Effect on Swelling Ratio (SR) and % Equilibrium swelling ratio (% EWS) of the NCHs

The study related to water absorption of CH‒guar gum/polyol and NCHs has shown interesting results. A detailed study of the water absorption behavior with the variation in the polyol content has been performed and presented in Appendix A; in brief, it shows that with the increase in the polyol content in the hydrogel matrix the swelling ratio also increased, which can be attributed to the higher crosslinking structure that formed, as is well supported by the FT-IR and SEM analyses. An SR of 9.432 for CGP2-5 was observed, which was quite a bit higher than that of our earlier synthesized hydrogels using synthetic MMA [36], as well as the natural polymer HEC [35]. The SR value in an acidic medium was also calculated for the CG-0, CGP2-3, and CGP2-5 hydrogels. This study revealed that the composition of CGP2-3 showed an SR value of 6.948 (Appendix A) in an acidic media of pH 4. A large variation in the water absorption in the various composition of hydrogels having different polyol contents was observed as the pH of the solution was varied. 

Noticeably, the presence of guar gum in the hydrogel matrix greatly influenced the swelling parameter. Guar gum, a galactomannan obtained from the Indian cluster bean *Cyamopsis tetragonoloba* (L.) Taub., is a water-soluble polysaccharide [44]. When it is mixed with chitosan and polyol, it gives rise to intermolecular hydrogen bonding and thus affects the swelling behavior. A literature review revealed that guar gum has been introduced for oral controlled drug delivery [45], and also for colon targeting [46]. The limitation associated with its high swelling capacity can be a factor in the early drug burst release phenomenon. Thus, the addition of hydrophobic polyol to the guar gum matrix will protect the drugs in the gastric liquid (pH 1.3–3.5) and the resulting hydrogels act as a site-specific delivery system, in which the degradation of guar gum component by the colonic microflora [47] may allow for the release of the drug at a niche where the hydrogel shows high swelling. The introduction of naturally occurring biodegradable polymers like chitosan, guar gum, and linseed oil-based polyol has led to environmentally friendly and commercially fruitful products. Furthermore, the increase in the SR and % EWS was studied for the various components of NCHs having different clay contents that are discussed below.

The equilibrium swelling behavior of the crosslinked NCHs hydrogels at 25 °C in pH 4 and pH 7.4 solutions is illustrated in Figure 9. The % EWS with the variation in pH of the NCHs constitutes an important parameter when designing a biomaterial for targeted DDSs. 

The phenomenon by which hydrogels swell in different pH allows a specific drug to be delivered to a specific site in the human body. It is well known that the basic principle on which hydrogel-based targeted drug delivery works upon is the pH-controlled swelling kinetics of the hydrogel network [48]. The clay content of the NCHs has a great influence on the water absorption capacity. The lower values of SR and % EWS of NC-4 and NC-6 can be mainly attributed to two factors.

Firstly, the hydrogen bond interaction between the polar groups present on the surface of the hydrophilic matrix, viz., the –OH and –CONH groups of chitosan, guar gum, and polyol with clay particles, can be weakened when it comes into contact with water, and the tangling of the polymeric chains was restricted [49]. Secondly, the presence of bulky alkyl chains may have restricted the complete radical polymerization reaction, decreasing the hydrophilicity and thereby the water absorption [50]. At pH 4.0, the SR ratio for the NC-2 containing 20% of Cloisite 30B was about 14.697 (Figure 10). The effect of Cloisite 30B content of NCHs on the SR value in pH 7.4 is shown in Figure 11. This SR value was far greater than that for a hydrogel without the clay, i.e., CGP2-1 to CGP2-5 (Appendix A). Similarly, excellent results were observed for the % EWS for NC-2 (Figure 9). A high % EWS value of 1369% for NC-2 can be explained on the basis of the classic rubber elasticity theory [51], which suggests that the total number of crosslinking points per clay platelet for NCHs is several tens to more than 100, which indicates that each clay particle acts as a multifunctional crosslinker; to a great extent, this minimizes the network voids for holding water and hence increases the crosslinking, providing a more porous structure well supported by the SEM micrographs (Figure 5). The excess of clay present in NC-4 and NC-6 may fill the voids and decrease the hydrophilic character in NCHs. 

#### 2.8.2. Swelling Studies in Physiological Fluids

An important relationship exists between the swelling of a polymeric hydrogel network and the nature of the solvent. Physiological solutions such as saline, urea, potassium nitrate, potassium iodide, D-glucose, and sodium hydroxide play an important role in exploiting the hydrogels in drug delivery. An effect of the cloisite 30B content in NCHs on the swelling ratio was also observed (Figure 11). Appendix A shows the % EWS of CGP2-5 and NC-6 in the above physiological solutions. The swelling study in 1% NaOH was also observed (Appendix A).

The swelling studies were carried out in irea (5% *w*/*v*), d-glucose (5% *w*/*v*), KI (15% *w*/*v*), NaCl (0.1%), KNO_3_, and KH_2_PO_4_ (the sources of K^+^) [37]. A high value of 1135.8% of % EWS was observed in the KH_2_PO_4_ solution (Figure 12). This is expected due to the different interactions of the pores of the hydrogel network with the ions of the salt. Notably, a 1000-fold higher % EWS than the initial dried weight was achieved in potassium iodide solution for the NCHs composition. 

### 2.9. Degradability Studies

#### 2.9.1. Hydrolytic Degradation 

The degradation behavior of the synthesized hydrogels in response to pH variation (4 vs. 7.4) was studied. It was observed that the gels containing a higher amount of polyol started their degradation within a week. The polymeric degradation mechanism can be understood in terms of chemical degradation via hydrolysis [52]. As the swelling process starts and the water molecule enters the hydrogel network, which contains hydrolyzable bonds, as soon as the equilibrium swelling is achieved and no further water can be absorbed by the matrix the degradation process is initiated and the chemical degradation results in a change in the basic structure of the polymeric network, giving rise to the release of monomers and oligomers. Moreover, the pH inside the pores is regulated by the acidic and basic pendant groups of the monomers as well as the pH of the solution in which the degradation is taking place. The hydrolytic degradation is measured as the % weight loss with time. The higher polyol content in CGP2-5 led to earlier degradation at pH 7.4 after EWS was achieved. This can be attributed to the ionization or protonation of the pendant groups. However, the gels with a higher content of polyol (CGP2-5) showed stability in acidic media. The hydrophobic nature of polyol and greater crosslinking density supported the stability of the hydrogels in acidic media. As the content of the ionized groups increases in the swelling medium, it becomes more hydrophilic, swells, and finally starts degrading. The composition of CGP2-7 having a comparably higher content of polyol due to this slow degradation of the hydrogels having an excess of polyol content showed a loosening of crosslinks and hence earlier degradation. The degradability studies on NCHs showed that the hydrogels containing a higher amount of Cloisite 30B exhibited earlier hydrolytic degradation at pH 4 after the EWS was achieved. This can be attributed to the deprotonation of the pendant groups. On the other hand, the NCHs containing the least Cloisite 30B took a longer time for degradation. The composition NC-2 was quite stable in acidic media, which can be attributed to the effective crosslinked structure, formed and hence resisting degradation for a longer period of time. Thus, the variation of polyol content and Cloisite 30B in the synthesized hydrogel system can be tuned in such a manner as to have tunable degradability kinetics and can be exploited to be used as sustained DDSs to prevent early drug release.

#### 2.9.2. Soil Burial Degradation

Biodegradation is the most essential parameter that needs to be optimized for a polymeric DDS. The need for control of degradation mechanisms for drug delivery devices has gained attention in recent years. The aim is to provide renal or fecal elimination [53], which results in minimum toxicity and enhances the therapeutic efficacy of the drug. In the present study, besides hydrolytic degradation, soil burial degradation was performed to see the complete degradability characteristics of the synthesized hydrogels [54]. All the hydrogels showed an increase in weight at the early stages of the soil burial degradation due to the enzymatic and microbial interactions. Fractures and surface degradation were clearly seen in the later stages of the study (Appendix A). The strength of the hydrogels containing a higher amount of polyol was affected.

## 3. Materials and Methods

### 3.1. Materials

Chitosan (448877-50G, Sigma Aldrich, St. Louis, MO, USA), guar gum (Product no. 39234, Acrylamide (AM), S.D. Fine-Chem limited, Mumbai, India), methylene bisacrylamide MBAAm (Product no. M2022-100G, Sigma Aldrich), tetra ethylene (TEMED) RM1572-100 mL, Himedia A.R., Mumbai, India), glacial acetic acid (Product no. 2105, Fisher Scientific, Pittsburgh, PA, USA), 50% hydrogen peroxide, diethyl ether, and acetic anhydride (Merck, Mumbai, India) were used as received. Linseed oil polyol was prepared by standard protocols [30]. Cloisite 30B was provided by Southern Clay (Gonzales, TX, USA) as a gift sample. Deionized water from Millipore (Mille U10 water purification system, Burlington, MA, USA) was used in the synthesis of hydrogels and their swelling studies. Sodium phosphate buffers of the required pH were prepared using standard commercially available buffer capsules ranging from pH 4.0 ± 0.05 to pH 7.0 ± 0.05. 

### 3.2. Methods

#### 3.2.1. Preparation of NCHs

CH‒GG‒polyol/Cloisite 30B hydrogel networks were prepared by free radical polymerization reaction using chitosan (2% by dissolving in 1% acetic acid) and guar gum (2%) as matrix material, and polyol (2%) mixture in different proportions. N,N′-methylenebisacrylamide (MBAAm) was used as a crosslinker, ammonium persulfate (APS) was used as an initiator, and N,N,N′,N′′-tetramethyl-ethylenediamine (TEMED) was employed as an accelerator. Firstly, the GG, CH, and polyol matrix is prepared, followed by the addition of water-dispersed Cloisite 30B. Two sets of compositions without Cloisite 30B (CH‒GG/polyol) (Table 1) and with the nanofiller Cloisite 30B (NCHs) (Table 2) were prepared. The gelation occurred at room temperature. Subsequently, the NCHs obtained were cut into 1 × 1 cm blocks using a surgical blade.

#### 3.2.2. Purification of the NCHs

The NCHs were washed with double-distilled water to eliminate any unreacted monomer and other moieties [36]. The NCHs were kept dispersed overnight in distilled water for complete purification. Next, these samples were dried at room temperature. 

#### 3.2.3. Fourier-Transform Infrared (FT-IR) Analysis of NCHs

The FT-IR analysis of the completely dried hydrogels was obtained using a model 1750 FT-IR spectrophotometer (Perkin Elmer Cetus Instruments, Norwalk, CT, USA).

#### 3.2.4. Thermal Gravimetric Analysis (TGA) of NCHs

The thermal stability of the synthesized hydrogels was analyzed using Perkin-Elmer Diamond analyzer (Shelton, CT, USA) under N_2_ at a heating rate of 10 °C/min. A hydrogel sample of about 10 mg was placed inside the hermitic aluminum lid. The temperature was increased from 100 to 500 °C. 

#### 3.2.5. Differential Scanning Calorimetry (DSC) of NCHs

DSC analysis was performed for using SII EXSTAR 6000, DSC620 (Tokyo, Japan) from 30 to 400 °C in an N_2_ atmosphere at a 10 °C/min heating rate.

#### 3.2.6. Scanning Electron Microscopy (SEM) Analysis of NCHs

The morphological analysis of dried and swollen samples in solutions of different pH (1 and 7.4) was done using a scanning electron microscope (LEO440, All India Institute of Medical Sciences, New Delhi, India ). The SEM analysis of swollen samples in different pH solutions was frozen in liquid N_2_, which helped to sustain the crosslinked structures without any rupture. These samples were dried in a lyophilizer and completely dried samples were fixed on the aluminum stubs with gold coating for 40 s.

#### 3.2.7. Transmission Electron Microscopy (TEM) of NCHs

TEM analysis was performed using JEOL-2100 TEM (Tokyo, Japan) at an operating voltage of 200 Kv. The samples were held using carbon-coated grids of a 300 mesh size. 

#### 3.2.8. X-ray Diffraction (XRD) of NCHs

The XRD analysis was done using a Rigaku MiniFlex diffractometer (using Cu Kα ¼ 1.5418 Å radiation, Tokyo, Japan) at a scanning rate of 0.002 per min in the range 30° to 80°.

#### 3.2.9. Swelling Studies

Swelling ratio (SR) and % equilibrium swelling ratio (% EWS) measurements were carried out using gravimetric measurements. The dried hydrogels were immersed in a solution of varying pH and 1% NaOH solution. The swollen hydrogels were weighed at regular intervals of time to monitor the swelling ratio and % EWS.

#### 3.2.10. Swelling Studies in Various Physiological Solutions

Swelling studies in various physiological solutions (saline, potassium nitrate, potassium iodide, d-glucose, and urea) were carried out by the standard gravimetric method [35,36].

#### 3.2.11. Hydrolytic Degradation

A hydrolytic degradation study was carried out using our method reported earlier [35,36]. 

#### 3.2.12. Soil Burial Degradation Studies

A soil burial degradation study was carried out by immersing a known weight of hydrogel samples in soil of a known composition as per the reported method [35,36]. 

## 4. Conclusions

This study highlights the use of the natural polymers chitosan and guar gum as matrix materials for the synthesis of biodegradable nanocomposite hydrogels, as evident from the hydrolytic and soil burial degradation studies. The role of linseed oil-based polyol and nanofiller Cloisite 30B in the fabrication of stable hydrogels was explored, with a view to introducing hydrophobicity and enhanced stability to physiological solutions. A three-dimensional network was formed, with the formation of hydrogen bonds giving rise to thermally stable gels. The complete exfoliation of the nanofiller resulted in highly swollen hydrogels. It is anticipated that the stable NCHs developed in this study have a hydrophobic property, high thermal stability, a pH-responsive nature, and enhanced swelling capacity, and so can find potential application as drug delivery systems, especially for the delivery of various hydrophobic drugs. We also hypothesize that this element of hydrophobicity may help these hydrogels to traverse the hydrophobic environment of the blood‒brain barrier and hence they may be explored as a potential vehicle for the delievery of various biologics to the brain. Future prospects for this hydrogel system lie in achieving a desired sustained-release system, with improved key features such as hydrophobicity, biodegradability, and pH-responsive swelling.

## Figures and Tables

**Figure 1 gels-05-00044-f001:**
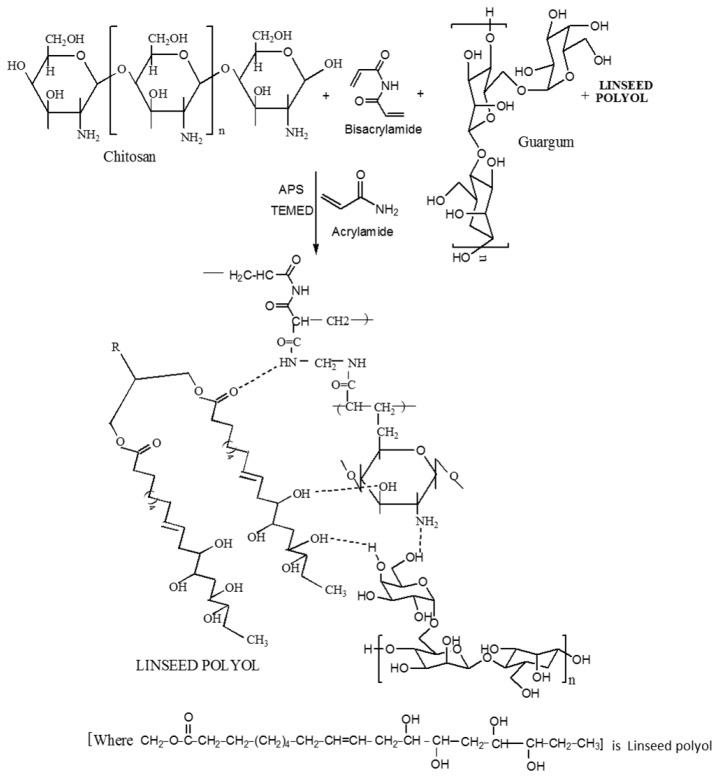
Reaction scheme showing the chemical synthesis of the interpenetrating network formation of CH‒guar gum (GG)‒polyol hydrogels.

**Figure 2 gels-05-00044-f002:**
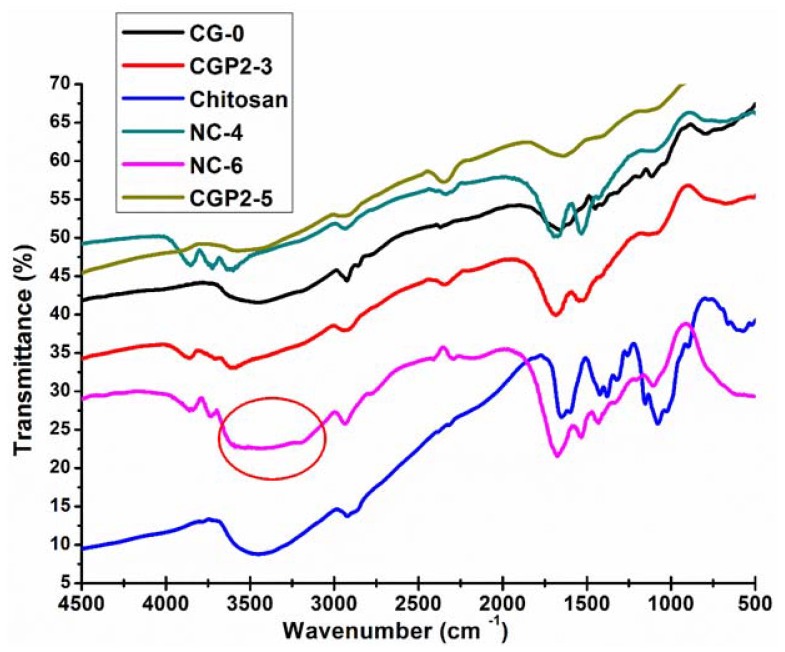
FT-IR spectra of pure chitosan, CG-0, CGP2-3, CGP2-5, NC-4, and NC-6 hydrogels.

**Figure 3 gels-05-00044-f003:**
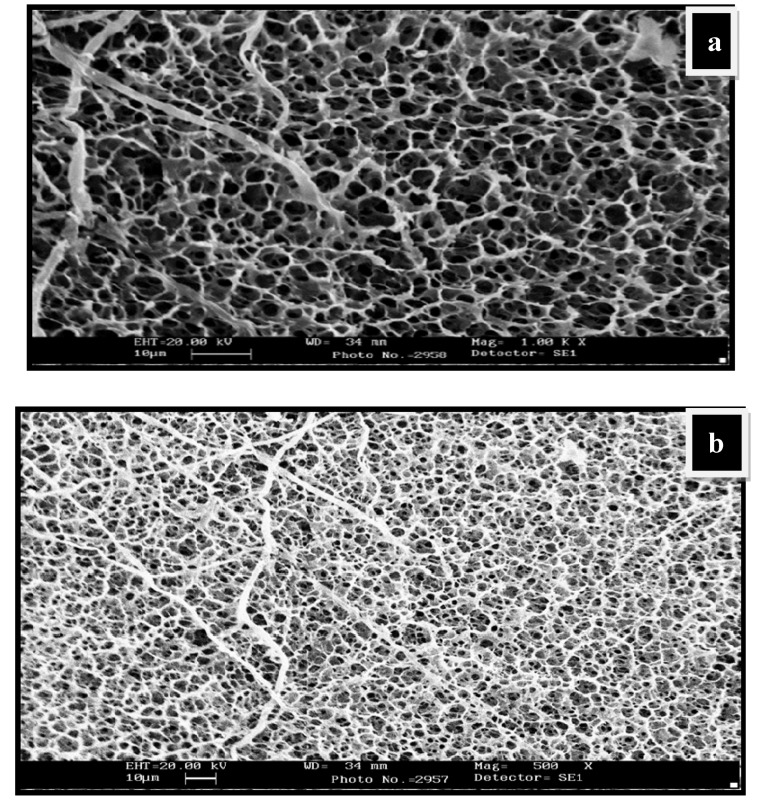
SEM micrographs showing water-swollen CG-0 hydrogel network: (**a**) Swollen structure at magnification 1.00K×; (**b**) swollen structure showing the interpenetrating network at magnification 500×.

**Figure 4 gels-05-00044-f004:**
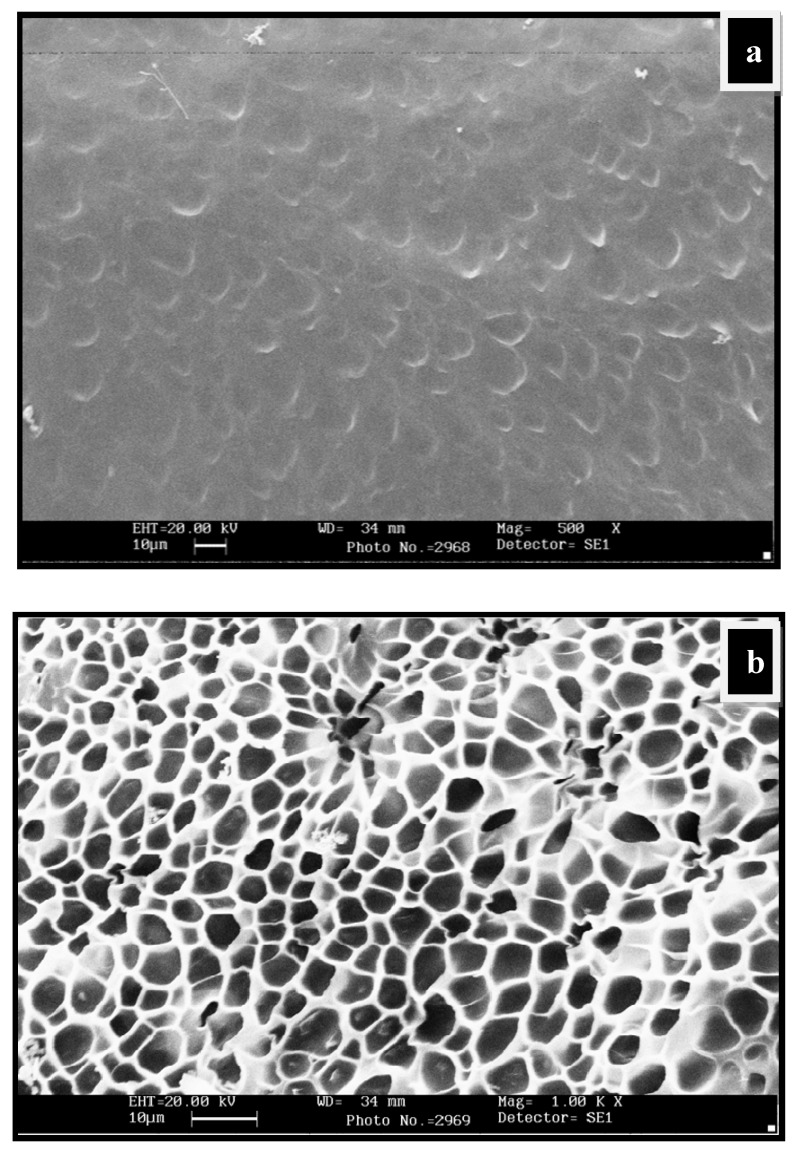
SEM micrographs of water-swollen CGP2-5: (**a**) Swollen structure at magnification 500×; (**b**) swollen structure showing the interpenetrating network at magnification 1.00K×.

**Figure 5 gels-05-00044-f005:**
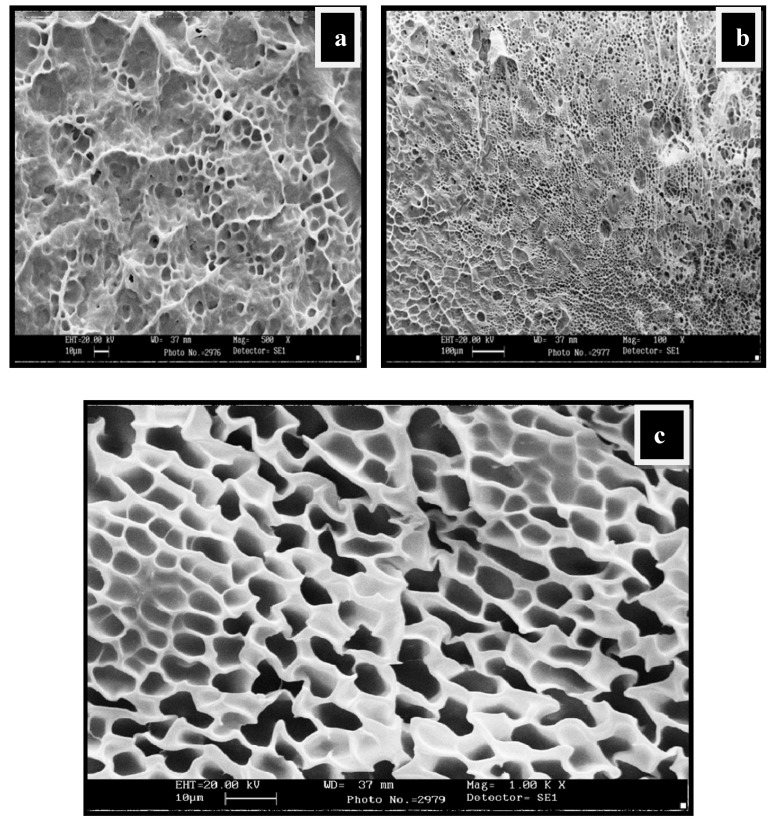
SEM micrographs showing water-swollen NC-2 hydrogel network: (**a**) Swollen structure at magnification 500K×; (**b**) swollen structure at magnification 100×; (**c**) swollen structure showing the interpenetrating network at magnification 1.00K×.

**Figure 6 gels-05-00044-f006:**
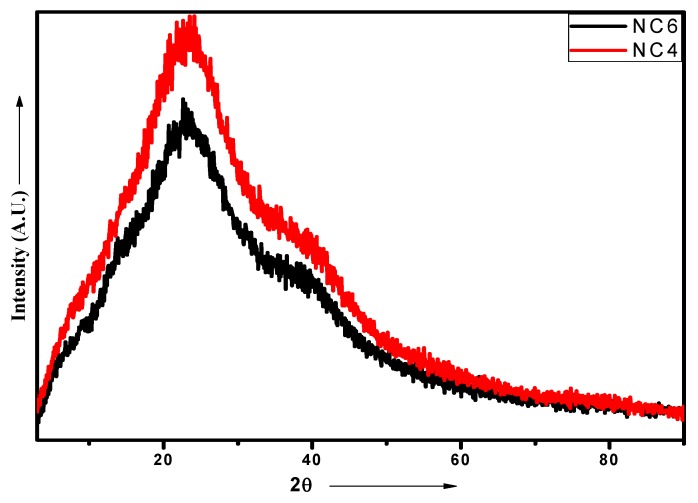
XRD patterns of NC-4 and NC-6 NCHs.

**Figure 7 gels-05-00044-f007:**
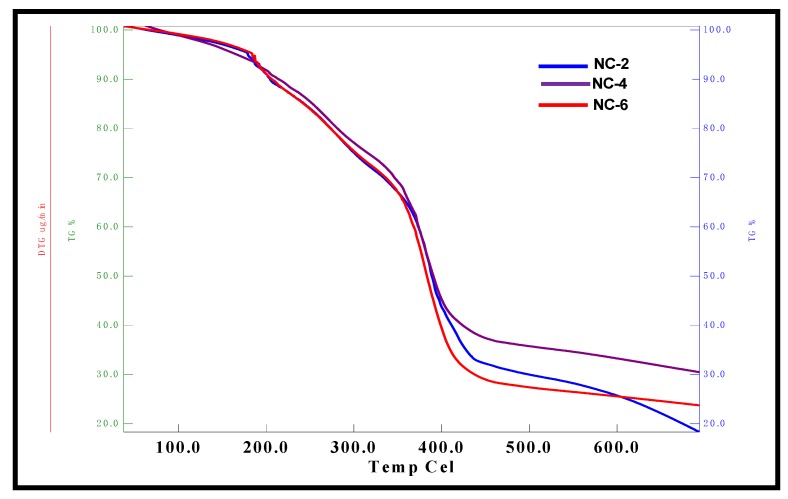
TGA micrographs of NC-2, NC-4, and NC-6 NCHs.

**Figure 8 gels-05-00044-f008:**
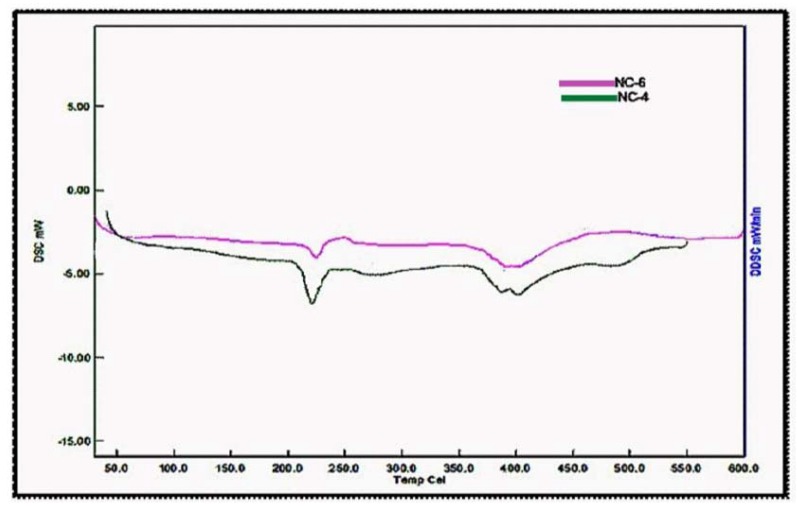
Differential scanning calorimetry micrographs of hybrid nanocomposites hydrogels (NCHs).

**Figure 9 gels-05-00044-f009:**
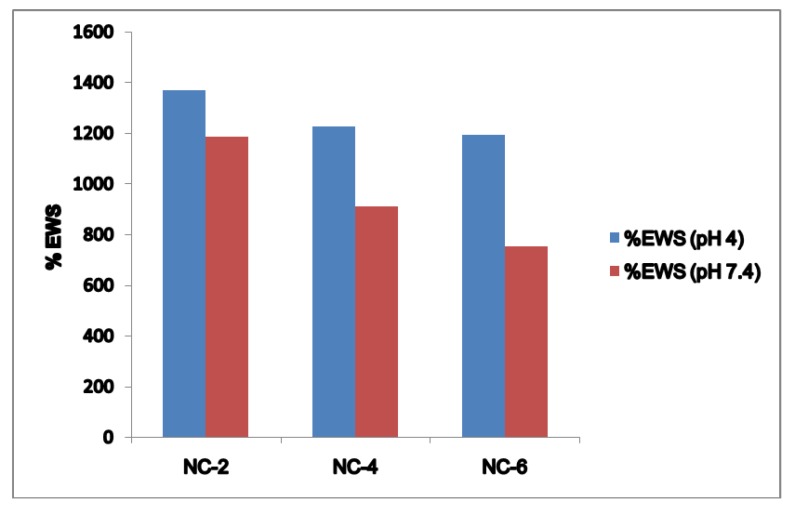
% EWS and pH relationship for the NCHs at pH 4 and pH 7.4.

**Figure 10 gels-05-00044-f010:**
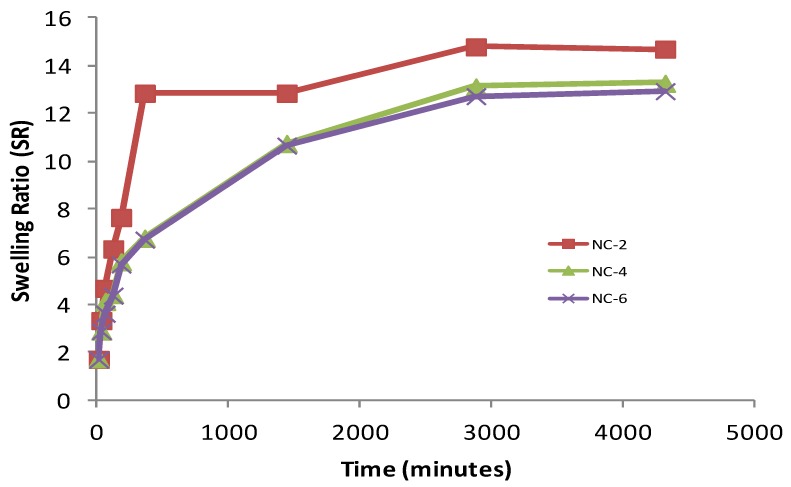
Effect of Cloisite 30B content on SR of various NCHs in phosphate-buffered saline, pH 4.0.

**Figure 11 gels-05-00044-f011:**
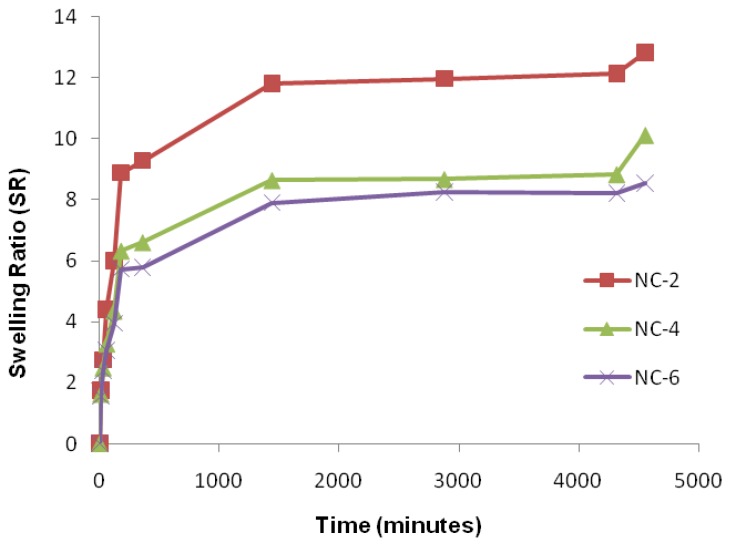
Effect of cloisite 30B content of NCHs on SR in phosphate-buffered saline, pH 7.4.

**Figure 12 gels-05-00044-f012:**
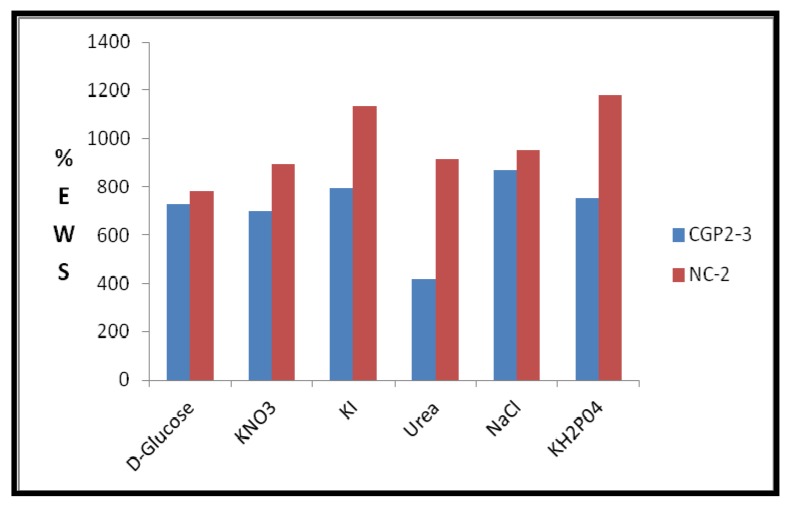
Effect of physiological solutions on % EWS of CGP2-3 and NC-2 hydrogels.

**Table 1 gels-05-00044-t001:** Feed compositions of the hydrogel network of CH‒GG/polyol.

S.No.	2% (*w*/*v*) Chitosan (mL)	2% (*w*/*v*) Guar gum (mL)	2% Polyol (mL)	AM:MBA (26:4) 30% (mL)	5% APS (mL)	TEMED (mL)
CG-0	20	20	0.0	10	1.0	0.1
CGP2-2	20	20	2.0	10	1.0	0.1
CGP2-3	20	20	3.0	10	1.0	0.1
CGP2-4	20	20	4.0	10	1.0	0.1
CGP2-5	20	20	5.0	10	1.0	0.1
CGP2-7	20	20	7.0	10	1.0	0.1

**Table 2 gels-05-00044-t002:** Feed compositions of NCHs.

S.No.	2% (*w*/*v*) Chitosan (mL)	2% (*w*/*v*) Guar gum (mL)	2% (*w*/*v*) Polyol (mL)	Cloisite 30B (mg/mL)	30% AM:MBA (26:4) (mL)	5% APS (mL)	TEMED (mL)
NC-2	20	20	3.0	2.0	10	1.0	0.1
NC-4	20	20	3.0	4.0	10	1.0	0.1
NC-6	20	20	3.0	6.0	10	1.0	0.1

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
