# Peer review of "Impact of Nanoclay on the pH-Responsiveness and Biodegradable Behavior of Biopolymer-Based Nanocomposite Hydrogels"

_gels, 2019, doi:10.3390/gels5040044_

Round 1

Reviewer 1 Report

This manuscript describes the fabrication of nanoclay/biopolymer-based nanocomposite hydrogels for potential applications in the biomedical field. 

Design and construction of functional hybrid hydrogels is one of the important fields of research and as a result, a range of nanocomposite hydrogels have been prepared for various applications. The introduction section does not provide an overview of the rapidly growing field. Further, synthesis and characterisation sections of this paper require significant improvement. For instance, Figure 3 does not contribute to the main text and need to be moved to SI. Figure 4, the SEM micrographs show very interesting porous networks. Unfortunately, the Authors have not discussed this data in great detail explaining the plausible mechanism for the formation of the porous structure. 

How were the hydrogel sample were prepared for SEM analysis - were they frozen in liquid N2 prior to the microscopy? If so, this needs to be discussed as the ice crystals are known to form pores in the soft materials such as hydrogels.

Rheology is one of the important technique to study the mechanical properties of the hydrogels and therefore the Authors should consider the effect of cross-linking etc on the visco-elastic properties of the hydrogels. 

Overall, the manuscript requires significant revisions, data presentation and discussions etc. and therefore, can not be recommended for publication in its current form. 

Author Response

Reviewer 1

This manuscript describes the fabrication of nanoclay/biopolymer-based nanocomposite hydrogels for potential applications in the biomedical field. 

Design and construction of functional hybrid hydrogels is one of the important fields of research and as a result, a range of nanocomposite hydrogels have been prepared for various applications. Comment 1: The introduction section does not provide an overview of the rapidly growing field. Further, the synthesis and characterization sections of this paper require significant improvement.

Response: As suggested by the reviewer, we have improved the introduction section by giving an overview of the rapidly growing field. And also improved the synthesis and characterization section.

Comment 2:  For instance, Figure 3 does not contribute to the main text and need to be moved to SI.

Response 2: As suggested by the learned reviewer we have moved the Figure 3 to Supporting information.

Comment 3: Figure 4, the SEM micrographs show very interesting porous networks. Unfortunately, the Authors have not discussed this data in great detail explaining the plausible mechanism for the formation of the porous structure. 

Response 3: As suggested by the reviewer we have given due explanation now for the interesting porous structures attained in the formation of nanocomposite hydrogels. Polyol and Nanoclay have played a crucial role in forming the interpenetrating networks and the porous structure obtained in the present system.

Comment 4: How the hydrogel sample were prepared for SEM analysis - were they frozen in liquid N2 prior to the microscopy? If so, this needs to be discussed as the ice crystals are known to form pores in the soft materials such as hydrogels.

Response 4: Yes liquid nitrogen was used to freeze the porous structure and then further lyophilized so that it is dried and to know the exact porous structures formed in the nanocomposite hydrogels. We have cited a reference for this procedure which is opted for the SEM analysis of swollen samples so that the porous crosslinked structure is not ruptured

Comment 5: Rheology is one of the important technique to study the mechanical properties of the hydrogels and therefore the Authors should consider the effect of cross-linking etc. on the visco-elastic properties of the hydrogels. 

Response 5: As suggested by the learned reviewer, this is an ongoing project in which we are in process of doing the drug delivery application aspects of the nanocomposite hydrogels and we will also consider the effect of crosslinking on the visco-elastic properties of hydrogels in future studies.

Comment 6: Overall, the manuscript requires significant revisions, data presentation and discussions etc. and therefore, cannot be recommended for publication in its current form. 

Response: We hope after considering all the comments of all the reviewers we have revised the manuscript significantly and modified the introduction and discussion and now can be recommended for the publication.

Reviewer 2 Report

 The manuscript needs modifications

1.      The title of the article is vague and should be rewritten

2.      The ‘Abstract’ should be more factual.

3.      The novelty of the work!? The significance of the work is not clear

4.      Please specify the role of this system in drug delivery and increase the readability of the manuscript

5.      The Y-axis of FTIR spectra? Fully misleading

6.      SEM analysis of the hydrogel sample by thermal drying? Please check the procedure of drying of hydrogel for SEM

7.      Figure 3 is not at all suitable

8.      TEM is not showing any feature

9.      Microstructures are poor in the manuscript

10.  Future prospects should be mentioned in the Conclusions

11.  The manuscript needs more clarity

Author Response

The manuscript needs modifications

Comments 1.      The title of the article is vague and should be rewritten

Response 1:  The article title is changed and has been reframed for the precision as “Impact of Nanoclay on the pH responsiveness and biodegradable behavior of biopolymers based nanocomposite hydrogels”

Comment 2.     The ‘Abstract’ should be more factual.

Response 2:   As suggested by the reviewer, the abstract has been changed and made for factual.

Comment 3.      The novelty of the work!? The significance of the work is not clear

Response 3:  The hydrophobic modification of the nanocomposite hydrogels using a vegetable oil derived linseed polyol has been reported for the first time in the present system.  This hydrophobic modification using linseed polyol has played a significant role in the modulation of the characteristics properties of the developed nanocomposite hydrogels.  It is to be noted that for the first time it has been demonstrated that the minimal addition of polyol to the guar gum-based hydrogels have influenced the stability and characteristic features of the hydrogels networks formed.

Comment 4.      Please specify the role of this system in drug delivery and increase the readability of the manuscript

Response 4: The role of the system in drug delivery is the future application of the developed nanocomposite hydrogels. We have revised the manuscript by directing the drug delivery applications as future prospects in the conclusions.

Comment 5.      The Y-axis of FTIR spectra? Fully misleading

Response 5: The y-axis has been well elaborated as % transmittance as suggested by the learned reviewer.

Comment 6.      SEM analysis of the hydrogel sample by thermal drying? Please check the procedure of drying of hydrogel for SEM

Response 6: The SEM analysis is done by freezing the swollen structure in liquid nitrogen and followed by lyophilization and we have cited a reference for this procedure which is opted for the SEM analysis of swollen samples so that the porous crosslinked structure is not ruptured.

Comment 7.      Figure 3 is not at all suitable

Response:  As suggested by reviewer 1 also we have moved the figure to supporting information showing the dried morphology of the developed nanocomposite hydrogels.

Comment 8.      TEM is not showing any feature.

Response: To visualize the layered structure of cloisite 30B was the goal of the study. Improved TEM images can be done in the future. We are removing the TEM image. 

Comment 9.      Microstructures are poor in the manuscript.

Response 9:  As suggested by other reviewers also Microstructures (SEM IMAGES) obtained for dry samples have been moved to supporting information.

Comment 10.  Future prospects should be mentioned in the Conclusions.

Response 10: As suggested by the reviewers, future prospects have been mentioned in the conclusion.

Comment 11.  The manuscript needs more clarity.

Response: We have tried our best to make the manuscript more clear by changing important parts of abstract, introduction, methodology, results and conclusion.

Reviewer 3 Report

This manuscript showed ambiguous description of synthesis of NCH and CGP.

No description on the results of synthesis of NCH in results and discussion.

In addition, there is no study for drug release test for these materials even though these are claimed as DDS. 

Author Response

Comment 1: This manuscript showed ambiguous description of synthesis of NCH and CGP.

Response 1: The synthesis methodology description has been improved as suggested by the reviewer to clearly mention the synthesis of NCH and CGP.

Comment 2.  No description on the results of synthesis of NCH in results and discussion.

Response 2: As suggested by the reviewer we have now discussed the synthesis of NCH in results and discussion.

Comment 3: In addition, there is no study for drug release test for these materials even though these are claimed as DDS. 

Response3: We have highlighted the synthesis and characterization of the nanocomposite hydrogels and highlighted the effect of polyol and Nanoclay on the characteristic features of the hydrogels. However, the drug delivery has been proposed as the future prospective application of the presented system and our future plan is to test the developed hydrogels for drug loading and release applications.

Round 2

Reviewer 1 Report

In light of the reviewers' comments, the authors have revised the manuscript sufficiently and can be recommended for publication after minor correction - see below

Regarding comment 4 - It should be noted that the porous structure/network observed during SEM analysis is due to the ice-templating effect. During freezing, the water molecules within the hydrogel network forms ice crystals, which on lyophilisation leaves porous network in the samples (ice-templating effect - significant literature is available in this area). Therefore, the observed pore structure may not be inherent to the hydrogels and is due to the freezing and lyophilisation process. 

Author Response

Comments

Comment: In light of the reviewers' comments, the authors have revised the manuscript sufficiently and can be recommended for publication after minor correction - see below

Regarding comment 4 - It should be noted that the porous structure/network observed during SEM analysis is due to the ice-templating effect. During freezing, the water molecules within the hydrogel network forms ice crystals, which on lyophilisation leaves porous network in the samples (ice-templating effect - significant literature is available in this area). Therefore, the observed pore structure may not be inherent to the hydrogels and is due to the freezing and lyophilisation process. 

Response: We agree with this comment. The phenomenon of Ice templating resulted in a porous structure. In this case, also, we have first frozen the swollen hydrogels having water inside in liquid nitrogen and then lyophilized the samples, and the porous structure due to ice templating effect may have been observed in the SEM analysis. However, our studies highlighted the cross-linked network formed due to a combination of biopolymers and polyol. Our earlier studies also show that the optical images of swollen hydrogels form the crosslinked network and Porous structure. (Vashist, Arti, Y. K. Gupta, and Sharif Ahmad. "Interpenetrating biopolymer network-based hydrogels for an effective drug delivery system." Carbohydrate Polymers 87, no. 2 (2012): 1433-1439.)  We have now highlighted the porous structures as observed during SEM analysis, which can be explained on the basis of the ice templating effect.

Reviewer 2 Report

manuscript can be accepted in the present form 

Author Response

Comment: manuscript can be accepted in the present form 

Moderate English changes required.

Response: We appreciate that the learned reviewer has accepted our manuscript after revision. As suggested by the learned reviewer we have made moderate changes in the English language.